# Adrenal Gland Irradiation Causes Fatigue Accompanied by Reactive Changes in Cortisol Levels

**DOI:** 10.3390/jcm11051214

**Published:** 2022-02-24

**Authors:** Yu-Ming Huang, Chih-Wen Chi, Pao-Shu Wu, Hung-Chi Tai, Ming-Nan Chien, Yu-Jen Chen

**Affiliations:** 1Department of Radiation Oncology, MacKay Memorial Hospital, Taipei 104, Taiwan; vin301.tw@gmail.com (Y.-M.H.); will@mmh.org.tw (H.-C.T.); 2Department of Medicine, MacKay Medical College, New Taipei City 252, Taiwan; pw2136@gmail.com (P.-S.W.); chienmingnan@gmail.com (M.-N.C.); 3Department of Biomedical Imaging and Radiological Sciences, National Yang Ming Chiao Tung University, Taipei 112, Taiwan; 4Department of Medical Research, MacKay Memorial Hospital, Taipei 104, Taiwan; d48906003@yahoo.com.tw; 5Department of Pathology, MacKay Memorial Hospital, Taipei 104, Taiwan; 6Division of Endocrinology and Metabolism, Department of Internal Medicine, MacKay Memorial Hospital, Taipei 104, Taiwan; 7Department of Artificial Intelligence and Medical Application, MacKay Junior College of Medicine, Nursing, and Management, New Taipei City 252, Taiwan; 8Department of Medical Research, China Medical University Hospital, Taichung 404, Taiwan

**Keywords:** adrenal gland, cortisol, fatigue, hypothalamic–pituitary–adrenal axis, radiotherapy

## Abstract

Background: Incidental radiotherapy (RT) to the adrenal gland may have systemic effects. This study aimed to investigate the effects of adrenal RT on fatigue. Methods: BALB/c mice were surgically explored to identify the left adrenal gland and delivered intra-operative RT. The swimming endurance test was used for endurance assessment to represent fatigue. Plasma levels of stress hormones and histopathological features were examined. Three patients with inevitable RT to the adrenal gland were enrolled for the preliminary study. Serum levels of cortisol, aldosterone, and adrenocorticotropic hormone (ACTH) were measured before and after RT. Fatigue score by using the fatigue severity scale and RT dosimetric parameters were collected. Results: In the experimental mouse model, adrenal RT decreased baseline cortisol from 274.6 ± 37.8 to 193.6 ± 29.4 ng/mL (*p* = 0.007) and swimming endurance time from 3.7 ± 0.3 to 1.7 ± 0.6 min (*p* = 0.02). In histopathological assessment, the irradiated adrenal glands showed RT injury features in the adrenal cortex. In the enrolled patients, baseline cortisol significantly declined after RT. There were no significant differences in the levels of morning cortisol, aldosterone, and ACTH before and after RT. Conclusions: The RT dose distributed to the adrenal gland may correlate with unwanted adverse effects, including fatigue and adrenal hormone alterations.

## 1. Introduction

Fatigue is one of the most common adverse effects of cancer and cancer treatment [1,2]. It is a complicated syndrome associated with multiple factors, such as progressive disease, anemia, pain, depression, sleep disturbances, poor nutrition, hormonal changes, and treatments including chemotherapy and radiotherapy (RT) [3,4,5]. The prevalence of fatigue in cancer patients, known as cancer-related fatigue, ranges from 25% to 99% in different reports [6]. After successful cancer treatment, fatigue remains problematic in 19% to 38% of patients [7].

Among the etiologies causing cancer-related fatigue, we observe a trend in clinical practice of the development of fatigue after incidental ionizing radiation to the adrenal gland. This kind of fatigue may not resolve after rest and requires extended recovery [8]. Indeed, there are various clinical situations with incidental adrenal radiation in the era of dose-painting RT, such as RT for hepatocellular carcinoma (HCC), liver metastasis, portal vein thrombosis, vertebral metastasis, and para-aortic lymph node metastasis. However, the correlation between adrenal ionizing radiation (AIR) and fatigue remains unclear.

The adrenal glands consist of the outer connective tissue capsule, the cortex, and the medulla. The adrenal cortex and medulla have distinct embryologic and physiological functions [9]. The adrenal cortex is composed of three zones—the zona glomerulosa, zona fasciculata, and zona reticularis—which release mineralocorticoids, glucocorticoids, and androgens, respectively. The zona fasciculata constitutes 75% of the adrenal cortex [10].

The essential glucocorticoids are cortisone and cortisol. Cortisone is the precursor of cortisol. Cortisol, the active form of glucocorticoids, has a half-life of approximately 3 h, whereas cortisone has a half-life of only half an hour [11]. Cortisol is involved in increasing blood sugar levels, immune responses, protein synthesis, and the transformation of glucose into glycogen [12,13]. Low levels of cortisol are related to fatigue due to its role in energy metabolism [14]. Cortisol is a stress hormone that is secreted in response to stress, such as fever, surgery, hypoglycemia, hypotension, and exercise [15,16]. Corticosterone is considered the main glucocorticoid involved in the regulation of stress responses in rodents. However, several studies have observed increased cortisol in plasma and adrenal glands of mice following stress [17,18,19,20,21], and many studies have used cortisol as the index for stress activation in mice [17,22,23,24,25].

The hypothalamic–pituitary–adrenal (HPA) axis is a psychoneuroendocrine regulator of the stress response and the immune system. Adrenocorticotropic hormone (ACTH) stimulates the secretion of glucocorticoids from the adrenal cortex. ACTH is secreted from the anterior pituitary under the influence of corticotropin-releasing hormone (CRH) and arginine vasopressin. CRH secretion is regulated by circadian rhythms and additional stressors operating through the hypothalamus. The secretion of CRH and ACTH is inhibited by cortisol, highlighting the importance of negative feedback control [26]. ACTH and cortisol are secreted with a circadian rhythm, with levels highest upon awakening and then declining [27,28]. In humans, cortisol levels peak in the morning with a nadir at night. Mice also display a circadian rhythm but peaking at night and with a nadir in the morning [17]. This study aims to investigate the effects of adrenal RT on fatigue and related physiological impacts in experimental animal models and clinical observation.

## 2. Materials and Methods

### 2.1. Experimental Animal Model

#### 2.1.1. Experimental Animals

Four-week-old male BALB/c mice were purchased from the National Laboratory Animal Center, Taiwan. Mice were maintained at a temperature of 22 ± 1 °C and humidity of 55 ± 10% with 12 hours of light (artificial illumination; 07:00–19:00). The mice were fed a commercial rodent diet (LabDiet 5001, PMI Nutrition International LLC, Brentwood, MO, USA) and allowed access to purified water in a water bottle.

After 1 week of pre-feeding, the mice were randomly grouped into 3 groups of 4 mice per group as follows: the sham group, the 0 Gy group, and the 2 Gy group. The mice in the sham group only had fur shaved around the left flank area. The mice in the 0 Gy group were surgically explored for identification of the left adrenal gland after fur shaving. The mice in the 2 Gy group received a radiation dose of 2 Gy on the left adrenal gland after surgical exploration and fur shaving.

After grouping, the blood of all mice was collected at 15:00 and 20:00 for the measurement of plasma levels of stress hormones. The cortisol levels at 15:00 were considered the baseline cortisol levels according to the circadian rhythm. Surgery and radiation were arranged on the next day (Figure 1a–c). The swimming endurance test was performed for exhaustion assessment to represent fatigue after wound healing around 5 days after surgery and RT. Then, another blood sampling was performed at 15:00 and 20:00 the next day. Histopathology of the adrenal gland was sampled 7 days after surgery and RT.

All procedures were approved by the Experimental Animal Committee at MacKay Memorial Hospital (MMH-A-S-103-06), in compliance with the National Institute of Health Guide for Care and Use of Laboratory Animals (NIH Publication, 8th edition, 2011). All efforts were made to minimize animal suffering, to reduce the number of animals used, and to utilize alternatives to in vivo techniques.

#### 2.1.2. Blood Analysis

Blood samples were collected via retro-orbital blood sampling and detected using Drew Hemavet HV950 (Drew Scientific, Inc., Dallas, TX, USA). Blood samples were centrifuged (1000× *g* for 15 min at 4 °C), and the serum was stored at −80 °C until use. Serum cortisol levels were quantified using enzyme-linked immunosorbent assay (ELISA) kits from Arbor Assays (Ann Arbor, MI, USA).

#### 2.1.3. RT Technique

After surgical exploration, intraoperative RT was administered to the left adrenal glands of the mice using a linear accelerator (IX, Varian, Palo Alto, CA, USA) with an electron beam (6 MeV, 90%, dose rate: 400 MU/min).

#### 2.1.4. Swimming Endurance Test

The swimming ability of all mice was confirmed using the swimming test before the experiment. The swimming test was stratified according to studies performed by Matsumoto et al. An acrylic pool (90 × 45 × 45 cm) filled with water to a depth of 38 cm was used. The temperature of the water was maintained at 25 ± 0.5 °C with a water heater and thermostat. The mice were allowed to swim until they failed to rise to the surface of the water to breathe within 7 s. The mice were rescued at the time to avoid drowning. For the swimming endurance test, all mice were loaded with lead wire weighted to 7% of body weight. These mice were then placed in a water tank for swimming (Figure 1d). The duration of starting swimming to fail to rise within 7 s was recorded as the swimming endurance time [17,29].

#### 2.1.5. Percentage Change (Δ)

The percentage changes (Δ) in baseline cortisol and evening cortisol before and after the experiment were recorded among the groups. There were no significant differences in Δ baseline cortisol, Δ evening cortisol, and the swimming endurance time between the sham group and the 0 Gy group (Table 1).

#### 2.1.6. Histopathology

Formalin-fixed and paraffin-embedded adrenal glands from each group of mice were sectioned. Sections 4 µm thick were stained with hematoxylin and eosin (H&E).

### 2.2. Clinical Observation—Preliminary Data

#### 2.2.1. Patients

Three patients with inevitable RT to the adrenal gland after standard optimization for intensity-modulated radiation therapy (IMRT) were prospectively enrolled. The patients were required to have a World Health Organization performance status of 0 to 2. All patients were treated with curative intent.

#### 2.2.2. Blood Analysis

Serum levels of cortisol at 08:00 and 16:00, aldosterone, and ACTH were measured on the day before RT, in the middle of the RT course, and 1 week after RT using radioimmunoassay kits. The cortisol levels at 16:00 were considered baseline cortisol levels according to the circadian rhythm.

#### 2.2.3. RT Technique

Patients underwent computed tomography (CT) simulation in the supine position and were immobilized with an alpha cradle. The adrenal glands were contoured on CT images while planning. The adrenal glands were located superior and anteromedial to the upper pole of the kidneys and appeared as triangular or y-shaped organs. IMRT was used for all patients. The treatment plans were generated using 6 MV or 10 MV photons. All patients were treated with linear accelerators, and dosimetric parameters, including the mean dose of the left adrenal gland, were collected using RT planning systems (Eclipse Treatment Planning System v.13; Varian Medical Systems Inc., Palo Alto, CA, USA). Dose distributions for the planning and dose–volume histograms (DVHs) were recorded for evaluation (Figure 2).

#### 2.2.4. Fatigue Severity Scale

Fatigue was scored according to the validated fatigue severity scale (FSS), a short questionnaire for rating the levels of fatigue on the day before RT and 1 week after RT [30,31]. The questionnaire contained 9 statements that attempted to explore the severity of fatigue symptoms. Patients were required to circle a number from 1 to 7, depending on how appropriate they felt the statement applied to them during the past week. A low value indicated that the statement was not applicable, whereas a high value indicated agreement. The scoring was performed by calculating the average response to the questions. Patients with fatigue reported an averaged high value.

#### 2.2.5. Ethical Statement

This study was approved by the Institutional Review Board of MacKay Memorial Hospital with ethics committee approval and informed consent (IRB number: 20MMHIS132e).

### 2.3. Statistical Analysis

Statistical analysis was performed using SigmaPlot version 12.0 (Systat Software, Inc., Chicago, CA, USA). Numerical data were expressed as mean ± standard deviation. A paired *t*-test was performed for serum levels of cortisol in mice, which were measured before and after RT, and FSS, serum levels of cortisol, aldosterone, and ACTH before and after RT for the human data. The Δ baseline cortisol, Δ evening cortisol, and the swimming endurance time of different groups of mice were recorded, and the significances were analyzed using independent *t*-test. Differences were considered significant if *p* < 0.05.

## 3. Results

### 3.1. Experimental Animal Model

#### 3.1.1. Blood Analysis

The baseline cortisol levels in the RT 2 Gy group significantly decreased from 274.6 ± 37.8 to 193.6 ± 29.4 ng/mL (*p* = 0.007), as indicated by paired t-test analysis (Figure 3a). The evening cortisol levels in the 2 Gy group decreased from 275.5 ± 39.4 to 217.5 ± 33.2 ng/mL (*p* = 0.05) with borderline significance (Figure 3b). The circadian rhythm was not apparent in both groups before or after the experiment (Figure 4). There was a significant difference in Δ baseline cortisol between the 0 Gy group and the 2 Gy group (*p* = 0.04), whereas no significant difference was noted in Δ evening cortisol between the 0 Gy and 2 Gy groups (*p* = 0.35) (Table 2) (Figure 5). There was no significant difference in hemoglobin, white blood cell counts, or platelets among the different groups of mice.

#### 3.1.2. Swimming Endurance Time

The swimming endurance time declined following RT, as 3.7 ± 0.3 and 1.7 ± 0.6 min in the 0 Gy and 2 Gy groups. There was a statistically significant difference in the 0 Gy and 2 Gy groups (*p* = 0.02), as indicated by independent *t*-test (Table 2) (Figure 6). An additional analysis was performed and showed that there were significant differences in the Δ evening cortisol levels (*p* = 0.01) and performance in the swimming endurance test (*p* = 0.03) between the sham group and 2 Gy group of mice.

#### 3.1.3. Histopathology

In the histopathological assessment, the irradiated adrenal glands showed RT injury features in the adrenal cortex, including moderate hypertrophy, disorganization, cellular aggregates, increased vasculogenesis, condensed chromatin in the nucleus, and cytoplasmic swelling, which were most prominent in the zona fasciculata (Figure 7). Histopathology of the unirradiated right adrenal cortex and RT-injured left adrenal cortex from each mouse of the 2 Gy group was illustrated in Appendix A.

### 3.2. Clinical Observation—Preliminary Data

#### 3.2.1. Patient Characteristics

A summary of the baseline characteristics of the three patients is listed in Table 3. All patients were male. Two patients were diagnosed as having esophageal cancer, and the other patient was diagnosed as having HCC. The median age at diagnosis was 52 years old (range 31–58 years old). One patient with esophageal cancer received neoadjuvant concurrent chemoradiotherapy (CCRT), and the other one received adjuvant CCRT after surgery. The patient with HCC received adjuvant RT for diaphragm and right adrenal gland invasion after segmental hepatectomy and right adrenal gland resection.

#### 3.2.2. RT Dosimetry

In the enrolled patients, the average volume of the unilateral adrenal gland was 3.2 ± 1.2 cm^3^. The mean RT dose to the unilateral adrenal gland was 33.6 ± 11.8 Gy.

#### 3.2.3. Blood Analysis

The baseline serum cortisol levels declined from 10.4 ± 2.7 (pre-RT) to 5.8 ± 2.8 (interim RT) and 4.6 ± 1.5 (post-RT 1 week) μg/dL. The morning cortisol levels declined from 14.8 ± 1.6 (pre-RT) to 9.4 ± 2.5 (interim RT) and 10.2 ± 1.3 (post-RT 1 week) μg/dL (Figure 8a). The aldosterone levels declined from 11.5 ± 2.4 to 8.5 ± 2.6 ng/dL (Figure 8b). The ACTH levels increased from 86.5 ± 42.1 to 98.9 ± 42.3 pg/mL (Figure 8c). The baseline cortisol levels significantly declined after RT, as indicated by paired t-test analysis (*p* = 0.002). Alterations in the levels of morning cortisol, aldosterone, and ACTH showed no significant differences. Before RT, the difference between morning and baseline cortisol levels was insignificant (*p* = 0.62). However, there was a significant difference between morning and baseline cortisol levels after RT (*p* = 0.02). It seemed that the circadian rhythm presented more prominent after RT (Figure 9).

#### 3.2.4. Fatigue Severity Scale

The FSS score slightly increased from 1.2 ± 0.1 to 3.1 ± 0.8 after RT without significant differences (Figure 10).

## 4. Discussion

In this study, we intended to illustrate the correlation between AIR and fatigue. Fatigue after AIR was observed with decreased cortisol. Further histopathology data showed RT injury features in the irradiated adrenal glands.

There was a relative lack of data regarding AIR-induced biological effects, which have only been mentioned in some case series. Casamassima et al. proposed the experience of stereotactic body radiation therapy (SBRT) for adrenal gland metastases at the University of Florence. Adrenal insufficiency after SBRT was documented as grade II in 1 of 48 patients [32]. Rubra et al. reviewed a series of 10 patients with adrenal metastases treated by SBRT at the University of Chicago. One patient developed grade II adrenal insufficiency 2.5 years after the completion of SBRT [33]. Wardak et al. reported one patient with bilateral adrenal gland metastases, which were all treated with SBRT. Blood analysis showed significantly increased ACTH levels and decreased cortisol levels, which raised concerns about primary adrenal insufficiency [34]. Chance et al. documented 43 patients with 49 adrenal metastases treated by SBRT. At a median dose of 60 Gy in 10 fractions, 50% of patients with bilaterally treated adrenal glands developed low-grade adrenal insufficiency after SBRT to adrenal metastases without acute high-grade toxicity [35].

Yuan et al. retrospectively analyzed 81 patients who received RT for adrenal metastases from HCC. The median radiation dose was 50 Gy, with a median fraction size of 2 Gy. Twenty-three patients reported grade I fatigue, and six patients reported grade II fatigue [36]. Mohnike et al. performed RT on adrenal gland metastases with interstitial high-dose-rate brachytherapy in 37 patients. The median biological equivalent dose (BED) was 37.4 Gy. Grade I or II toxicities occurred in 11 patients (29%), including pain, nausea, vomiting, and fatigue. One grade 3 event (bleeding) occurred (3%). Ongoing cortisone substitution after treatment was required in two patients, while one patient required intermittent cortisone substitution for 1 month post-treatment [37]. König et al. evaluated 28 patients administered SBRT for adrenal gland metastases. The median BED was 75 Gy. The most common acute side effect was fatigue, two patients with grade I, and four patients with grade II. Fatigue as late toxicity occurred in two patients, one patient with grade I, and the other one with grade II [38].

Various multicenter series on patients who received RT to adrenal glands have been published in recent years. Some series reported detailed toxicity profiles, including adrenal insufficiency after one-sided or two-sided RT. Most radiation-induced adrenal toxicities were grade I or II [39]. There was sparse literature considering the adrenal glands as organs at risk in RT. The evaluation of toxicity profiles was mostly from SBRT for adrenal metastases [40,41,42]. We regarded healthy adrenal glands, but not metastatic ones, as essential organs to spare in RT planning, which differed from previous studies. Therefore, the toxicities should be the lower the better for the incidental RT dose to the adrenal gland. AIR might be related to adrenal insufficiency or fatigue in previous studies. However, no previous clinical data addressed the direct relationship of AIR, adrenal insufficiency, and fatigue, and monitoring of the HPA axis was not performed or was incomplete in most studies. We developed an experimental animal model for mice AIR with histological proof of the effects of radiation and cortisol. Furthermore, there was full consideration of the HPA axis and circadian rhythm in the blood analysis in our clinical data, and FSS was used for patients to respond to their extent of fatigue.

In addition to the direct RT damage on the adrenal gland, localized RT may produce systemic effects via inflammatory responses. There were several studies in the literature discussing about systemic tumor necrosis factor alpha (TNF-α) induced by localized RT in normal tissues. The release of intracellular molecules from RT-induced injured cells would trigger innate immune responses characterized by the upregulation of pro-inflammatory cytokines such as TNF-α [43,44,45]. Previous studies have shown that TNF-α inhibited basal and ACTH-stimulated cortisol secretion [46,47]. Additionally, RT-induced fatigue was associated with increased pro-inflammatory gene expression in normal tissues such as liver TNF-α mRNA and in circulation [43]. Therefore, RT effects on non-adrenal tissue might influence adrenal function, and should be examined in further studies.

Surgical series indicated that after complete removal of one adrenal gland, only a minority of patients developed clinically relevant adrenal insufficiency [48]. The extent, timing, and duration of clinical symptoms and hormone alterations induced by surgery and adrenal RT might be different, and regulation of the HPA axis could be distinct. In our study, RT-induced adrenal insufficiency was observed. Further research would be required for this issue, and RT-induced direct adrenal damages and TNF-α production that inhibited basal and ACTH-stimulated cortisol secretion may be possible causes.

Chemotherapy may influence the study results. Cisplatin has been reported to induce corticosteroid release in an animal model [49] and it has been reported to be associated with decreased cortisol levels in human [50]. There were several studies in the literature discussing about chemotherapy-induced adrenal insufficiency in cancer patients [51,52]. For patients treated with CCRT, adrenal insufficiency might be attributed to RT and chemotherapy. The impacts of RT and chemotherapy should be evaluated separately, and may need further investigation.

Corticosteroids have been widely used with chemotherapy for prophylaxis and treatment of chemotherapy-related symptoms, and may influence adrenal function [53]. The impact of corticosteroids on adrenal function has been proved to be dose- and duration-dependent. Use of corticosteroids in low or high doses resulted in a percentage of adrenal insufficiency of 2.4 and 21.5%, respectively, and short- or long-term use resulted in a percentage of adrenal insufficiency of 1.4% and 27.4%, respectively [54]. In this study, two patients received CCRT with a single low dose of corticosteroids (dexamethasone 4 mg via infusion) weekly prior to chemotherapy. If corticosteroids were administered at high doses, the serum cortisol levels might be suppressed within 24–48 h, recovering after 1–4 weeks [55]. Therefore, use of corticosteroids might have relevantly influenced the results, but the impact might be relatively unremarkable in short-term low-dose administration.

Other possible etiologies causing fatigue, such as electrolyte imbalance, infections, and mental state, remain to be excluded. In our study, all mice were fed sufficient food and purified water. No body weight loss was noted before and after the experiment. However, they lived in cages and possibly with the stress and bad memory of blood harvest after their first experience. We observed the mice twice a day (day and night), and decreased activity and pain were noticed only a few days after surgery and RT. The mice recovered soon without specific unhealthy signs such as changes in bowel habits or unhealed wounds. The swimming endurance test was then applied with heat lamps and blankets prepared. Despite these precautions, we could not confirm the health and infection statuses of the mice. When it came to humans, the etiologies of fatigue were much harder to evaluate due to different health statuses, socioeconomic statuses, family backgrounds, and personalities.

There were several limitations to our study. First, the causal relationship between AIR and fatigue, which was assessed by decreased swimming endurance time, remained undetermined and may require further investigation. Secondly, the number of patients in our clinical observation with assessable serum cortisol and ACTH levels was too small to draw a firm conclusion, and the number of mice in our animal study was also too small. Dose–response analysis may be performed if a larger study population is included. A third limitation was the timing and duration of AIR-induced biological effects may need a precise and long-term follow-up for evaluation in a large-scale study. Fourthly, a non-adrenal irradiation control group may help to examine the effects of RT to normal tissue on adrenal insufficiency. Fifthly, ACTH was not measured in the animal study to distinguish between central or peripheral adrenal insufficiency. The blood samples were collected via retro-orbital blood sampling at 15:00 and 20:00 on the sampling day to represent baseline and evening cortisol levels. For the little volume of each sampling, we only analyzed cortisol levels in this study. Finally, other possible etiologies such as chemotherapy that cause fatigue should be examined in future work.

This investigation is the first study in which AIR-induced biological effects have been verified by an experimental animal model. Our data show that low doses of ionizing radiation (single shot of 2 Gy) moderately impact swimming endurance time and cortisol levels. These novel findings provide critical information for clinical dose-painting RT, such as IMRT, SBRT, and intensity-modulated proton therapy (IMPT), with possibly ignored adverse effects on both fatigue and the HPA axis. The development of a new constraint for RT planning for the adrenal glands might be feasible.

## 5. Conclusions

Our preliminary clinical observations of fatigue development and the accompanying characteristic hormone alterations indicate that the adrenal glands could be regarded as an organ at risk from RT. This observation is supported by experimental animal models with functional and histopathological evidence. However, considering the small number of cases reviewed, a larger prospective study is required for future work, and further clinical investigations to validate our findings are warranted.

## Figures and Tables

**Figure 1 jcm-11-01214-f001:**
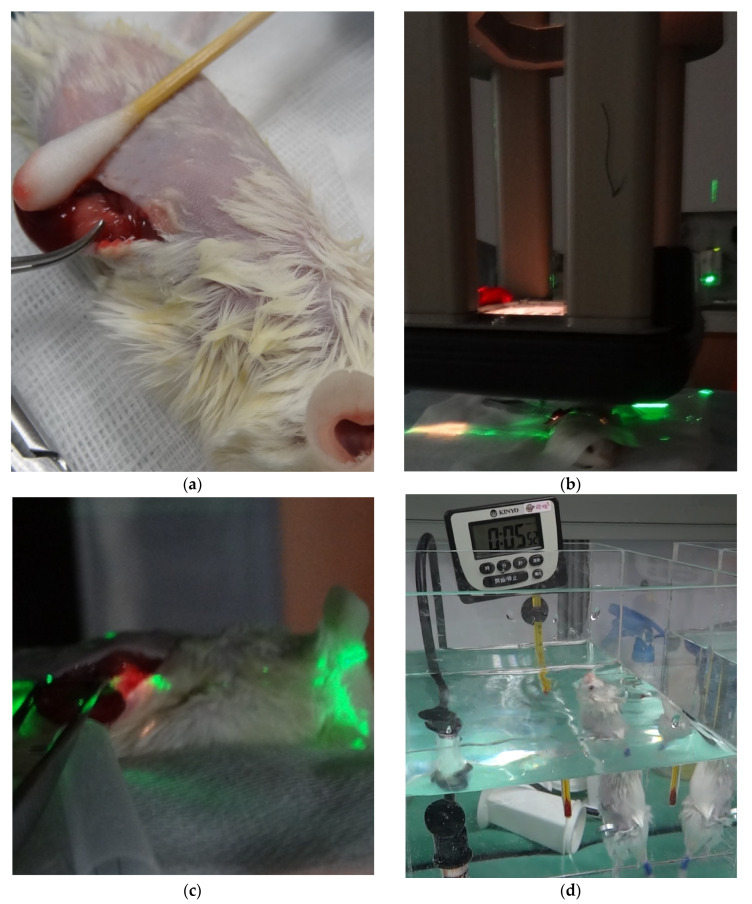
The experimental animal model. (**a**) The left adrenal gland was surgically explored and exposed to RT. (**b**) After positioning was confirmed by the laser system, intraoperative RT was administered to the left adrenal glands of mice using a linear accelerator with an electron beam. (**c**) A light field was used for adrenal gland localization. (**d**) All mice underwent a swimming test for endurance assessment as a representation of fatigue. Abbreviations: RT, radiotherapy.

**Figure 2 jcm-11-01214-f002:**
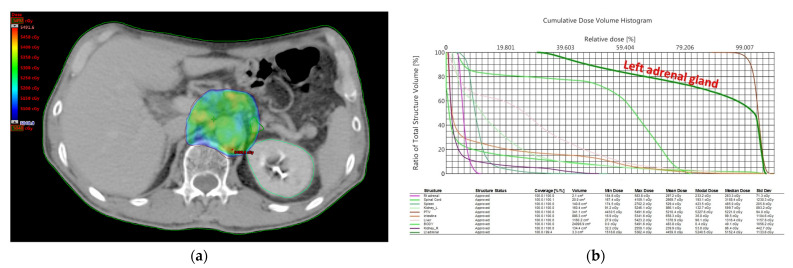
RT plan for one patient with a mean dose of 44.5 Gy to the left adrenal gland. (**a**) The y-shaped left adrenal gland was contoured with a dark green line in the planning CT images. (**b**) Dose distributions and DVHs were recorded for evaluation. The dose distribution of the left adrenal gland was outlined by a dark green line in cumulative DVH. Abbreviations: RT, radiotherapy; CT, computed tomography; DVH, dose–volume histogram.

**Figure 3 jcm-11-01214-f003:**
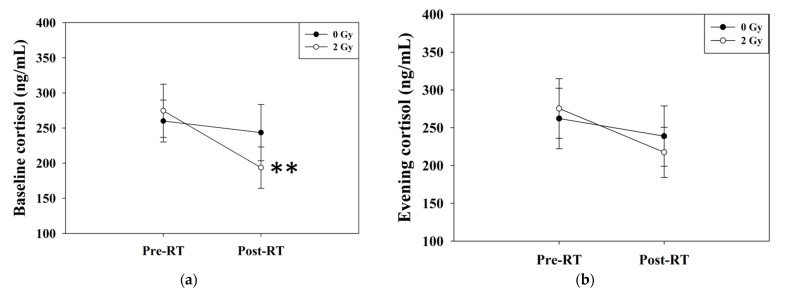
Serum levels of (**a**) baseline cortisol and (**b**) evening cortisol. The levels of baseline cortisol declined after RT (*p* = 0.007) by the paired *t*-test. The levels of evening cortisol in the 2 Gy group decreased with borderline significance (*p* = 0.05). ** Differences were significant at the 0.01 level (2-tailed). Abbreviations: RT, radiotherapy.

**Figure 4 jcm-11-01214-f004:**
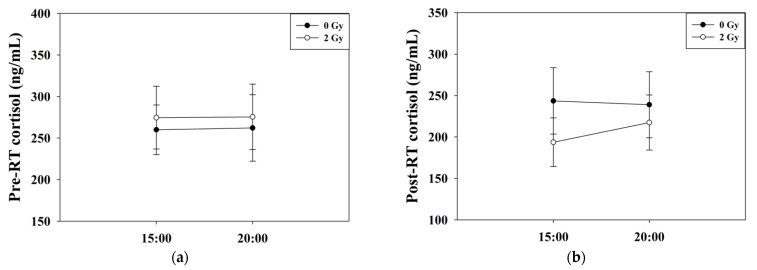
Circadian rhythms of cortisol levels. The circadian rhythm was not apparent (**a**) before and (**b**) after the experiment in both groups.

**Figure 5 jcm-11-01214-f005:**
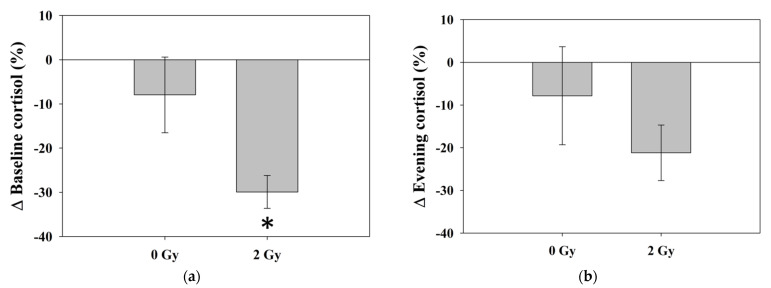
The percentage changes in (**a**) baseline cortisol and (**b**) evening cortisol of the 0 Gy and 2 Gy groups. There was a significant difference in Δ baseline cortisol between the 0 Gy group and the 2 Gy group (*p* = 0.04), whereas no significant difference was noted in Δ evening cortisol between the 0 Gy and 2 Gy groups (*p* = 0.35). * Differences were significant at the 0.05 level (2-tailed). Abbreviations: Δ, percentage change.

**Figure 6 jcm-11-01214-f006:**
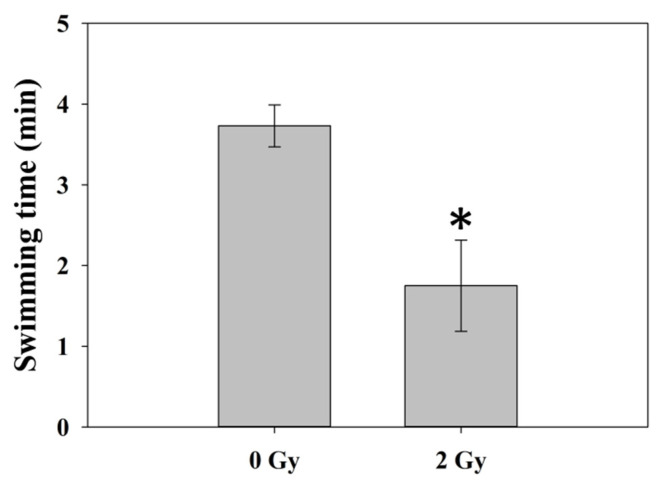
Swimming endurance time. The swimming endurance time decreased following RT. There was a statistically significant difference in 0 Gy and 2 Gy groups (*p* = 0.02) by independent *t*-test. * Differences were significant at the 0.05 level (2-tailed). Abbreviations: RT, radiotherapy.

**Figure 7 jcm-11-01214-f007:**
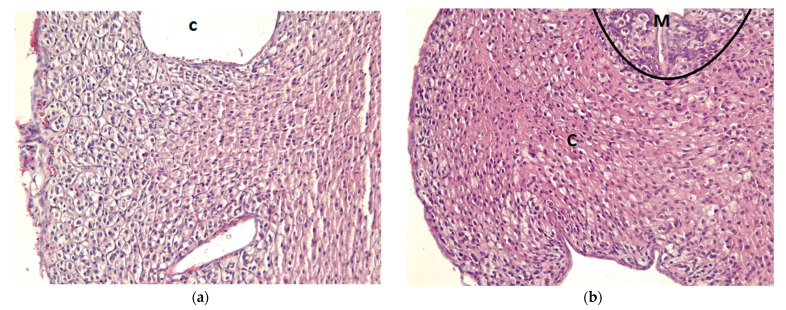
Histopathological changes induced by RT, as indicated by H&E staining. (**a**) Histopathology of the left adrenal gland from one of the 0 Gy group. (**b**) Histopathology of the left adrenal gland from one of the 2 Gy group. The irradiated adrenal glands showed RT injury features in the adrenal cortex, including moderate hypertrophy, disorganization, cellular aggregates, increased vasculogenesis, condensed chromatin in the nucleus, and cytoplasmic swelling, which were more prominent in the zona fasciculata. Abbreviations: C, cortex; M, medulla; RT, radiotherapy; H&E, hematoxylin and eosin.

**Figure 8 jcm-11-01214-f008:**
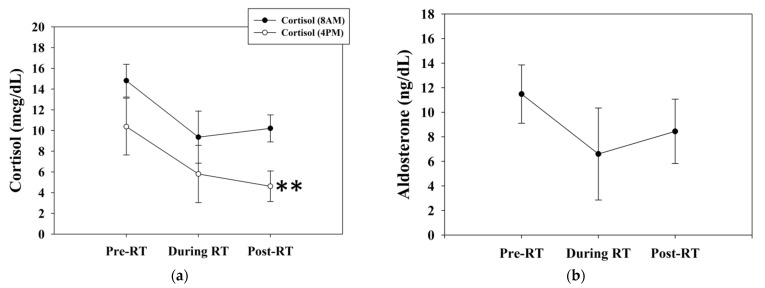
The serum levels of (**a**) morning cortisol, baseline cortisol, (**b**) aldosterone, and (**c**) ACTH. Baseline cortisol levels significantly declined after RT (*p* = 0.002). There were no significant differences in the levels of morning cortisol, aldosterone, and ACTH before and after RT. ** Differences were significant at the 0.01 level (2-tailed). Abbreviations: ACTH, adrenocorticotropic hormone; RT, radiotherapy.

**Figure 9 jcm-11-01214-f009:**
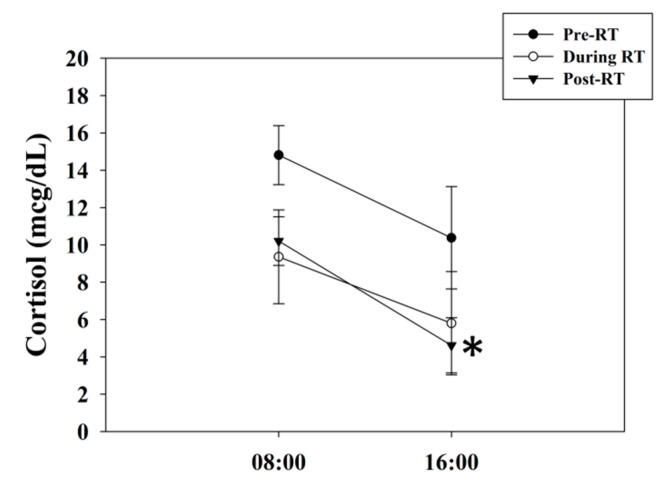
The effect of circadian rhythms on cortisol levels. The difference between morning and baseline cortisol levels was insignificant (*p* = 0.62) before RT. After RT, there was a significant difference between morning and baseline cortisol levels (*p* = 0.02). The circadian rhythm seemed to exert a greater effect after RT. * Differences were significant at the 0.05 level (2-tailed). Abbreviations: RT, radiotherapy.

**Figure 10 jcm-11-01214-f010:**
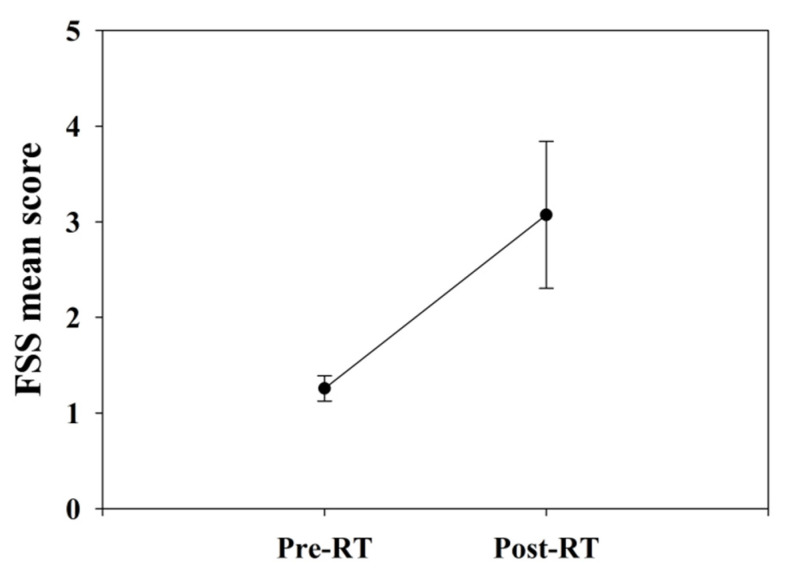
The fatigue severity scale of all patients. The FSS score slightly increased after RT without significant differences (*p* = 0.11). Abbreviations: FSS, fatigue severity scale; RT, radiotherapy.

**Table 1 jcm-11-01214-t001:** Δ Baseline and evening cortisol, endurance swimming time of sham and 0 Gy groups.

Variables	Sham	0 Gy	*p*
	M ± SD ^1^	M ± SD	
Δ Baseline cortisol (%)	−9.6 ± 7.8	−7.9 ± 8.5	0.89
Δ Evening cortisol (%)	11.4 ± 2.2	−7.8 ± 11.5	0.22
Endurance swimming time (min)	3.3 ± 0.1	3.7 ± 0.3	0.26

^1^ Numerical data were expressed as mean ± standard deviation.

**Table 2 jcm-11-01214-t002:** Δ Baseline and evening cortisol, endurance swimming time of 0 Gy and 2 Gy groups.

Variables	0 Gy	2 Gy	*p*
	M ± SD ^1^	M ± SD	
Δ Baseline cortisol (%)	−7.9 ± 8.5	−29.9 ± 3.7	0.04 *
Δ Evening cortisol (%)	−7.8 ± 11.5	−21.2 ± 6.5	0.35
Endurance swimming time (min)	3.7 ± 0.3	1.7 ± 0.6	0.02 *

^1^ Numerical data were expressed as mean ± standard deviation. * Differences were significant at the 0.05 level (2-tailed).

**Table 3 jcm-11-01214-t003:** Baseline characteristics of all patients ^1^.

Diagnosis	Age	PS ^2^	Stage	Pre-RT Treatment ^3^	Chemotherapy ^4^	Post-RT Treatment ^3^	Pre-RT Hb (g/dL)	Pre-RT Alb (g/dL)
Esophageal cancer	58	0	IIB	OP	Weekly CDDP (30)	-	12.6	3.5
Esophageal cancer	52	0	IIIC	-	Weekly CDDP (20) + Taxol (50)	OP	13.7	4
HCC	31	0	IIIC	OP	-	-	10.6	4.6

^1^ Abbreviations: PS, performance status; RT, radiotherapy; Hb, hemoglobin; Alb, albumin; OP, operation; CDDP, cisplatin; HCC, hepatocellular carcinoma; ECOG, Eastern Cooperative Oncology Group; CCRT, concurrent chemoradiotherapy. ^2^ The performance status was graded with ECOG score, in which grade 0 represented fully active. ^3^ Two patients with esophageal cancer received esophagectomy and lymph node dissection. The patient with HCC received adjuvant RT to the surgical bed for diaphragm and right adrenal gland invasion after segmental hepatectomy and right adrenal gland resection. ^4^ The patient with esophageal cancer treated with adjuvant CCRT received weekly cisplatin (30 mg/m^2^), and the other one treated with neoadjuvant CCRT received weekly cisplatin (20 mg/m^2^) and paclitaxel (50 mg/m^2^).

## Data Availability

Not applicable.

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
