# Peer review of "Adrenal Gland Irradiation Causes Fatigue Accompanied by Reactive Changes in Cortisol Levels"

_jcm, 2022, doi:10.3390/jcm11051214_

Round 1
Reviewer 1 Report
The authors present an interesting study on the association of post-irradiation fatigue with the post-irradiation changes in the adrenal gland function. The paper is written clearly, however, it needs English language and style editing - I suggest the authors ask a native English speaker to check the manuscript or use professional language editing services. The authors used proper methods to evaluate the changes in adrenal gland functions and fatigue after radiotherapy in the animal model and found significant changes between the groups without radiation therapy and the group that received radiotherapy. Moreover, they included three patients who had to undergo radiotherapy in the area of the adrenal glands and measured adrenal functions before and after the therapy was administered. Although radiotherapy is considered one of the causes of adrenal insufficiency, I was not able to find a similar study proving it.
My only question to the authors is if there were significant differences in the cortisol levels and performance in the swimming endurance test between the sham group and Gy2 group of mice.
Author Response
Thanks for your suggestion for this article. We have made a point-by-point responses as follows.
Comment of reviewer
- My only question to the authors is if there were significant differences in the cortisol levels and performance in the swimming endurance test between the sham group and 2 Gy group of mice.
Response
- There were significant differences in the Δ evening cortisol levels (p = 0.01) and performance in the swimming endurance test (p = 0.03) between the sham group and 2 Gy group of mice. The mice in the sham group only had fur shaved around the left flank area. The mice in the 0 Gy group were surgically explored for the identification of the left adrenal gland after fur shaving. The mice in the 2 Gy group received a radiation dose of 2 Gy on the left adrenal gland after surgical exploration and fur shaving. Concerning the influence of the operation on adrenal function while comparing the effect of RT, we initially only compared the 0 Gy group and 2 Gy group. To clarify the viewpoint from this comment, we performed an additional analysis and added an additional description in Results as follows: “An additional analysis was performed and showed that there were significant differences in the Δ evening cortisol levels (p = 0.01) and performance in the swimming endurance test (p = 0.03) between the sham group and 2 Gy group of mice.”

Reviewer 2 Report
Reviewer Comments to Author
Thank you for the chance to review this article in which the authors provide an interesting analysis of systemic affects of radiation therapy to the adrenal glands. The authors should be commended for exploring this subject which is of relevance for the readership of the journal, assuming radiation oncologists or radiation biologists are among the readers. There are several issues with the dataset and the interpretation; key points are as follows:
◦ Time frames: It is unclear how long irradiation of one adrenal gland would lead to depression of hormone levels until levels are compensated by the contralateral gland or remaining tissue on the ipsilateral gland.
◦ The experimental setup of the authors only allows for a short-term conclusion as the measurements were done five days after surgery. It is possible that the animals would quickly recover following the post-operative period. I suppose the data are unavailable due to the sampling of histopathology seven days after RT
◦ Surgical series indicate that even after complete removal of one gland, only a minority of patients develops clinically relevant adrenal insufficiency; therefore, it is unlikely that there is more damage caused by one-sided RT. This should be discussed (e.g., Mitchell et al., 2009, Surg Endosc)
◦ Multiple multicenter series on patients who received radiation therapy to adrenal glands have been published in recent years; some with detailed toxicity reports, including on adrenal insufficiency after one-sided or two-sided RT (e.g., Buergy et al., 2021). Although these series mostly reported on clinical findings (not laboratory studies); they indicate that clinically relevant symptoms are rare after one-sided adrenal gland RT
◦ Chemotherapy may have influenced the results (e.g., Cisplatin has been reported to induce Corticosteroid release in an animal model, Aggarwal et al., Anticancer Drugs 1994; furthermore, it has been reported to be associated with decreased cortisol levels after application in patients: Morrow et al., 2002, psychophysiology); whichever effect
◦ Corticosteroids which are typically applied in conjunction with chemotherapy and are used in many patients treated with RT may have influenced the results (Yeung et al., 1998, Endocrine Reviews); did all three patients receive corticosteroids to mitigate anticancer treatment effects or only the two (of three) chemotherapy patients?
It has to be acknowledged that the study provides relevant data despite the weaknesses discussed above; furthermore, it is likely impossible to address most of the points within a major revision. I therefore recommend that the authors thoroughly discuss aforementioned points. If further cortisol levels were taken in patients after the 1-week period, these data should be provided (i.e., if the authors have longer-term data, these would be valuable and should be included in the study to reduce the risk of short-term effects which are irrelevant over a longer time frame).
Author Response
Thanks for your suggestion for this article. We have made some point-by-point responses to the comments of reviewer.
Comments of reviewer
- Time frames: It is unclear how long irradiation of one adrenal gland would lead to depression of hormone levels until levels are compensated by the contralateral gland or remaining tissue on the ipsilateral gland.
- The experimental setup of the authors only allows for a short-term conclusion as the measurements were done five days after surgery. It is possible that the animals would quickly recover following the post-operative period. I suppose the data are unavailable due to the sampling of histopathology seven days after RT.
- Surgical series indicate that even after complete removal of one gland, only a minority of patients develops clinically relevant adrenal insufficiency; therefore, it is unlikely that there is more damage caused by one-sided RT. This should be discussed (e.g., Mitchell et al., 2009, Surg Endosc).
- Multiple multicenter series on patients who received radiation therapy to adrenal glands have been published in recent years; some with detailed toxicity reports, including on adrenal insufficiency after one-sided or two-sided RT (e.g., Buergy et al., 2021). Although these series mostly reported on clinical findings (not laboratory studies); they indicate that clinically relevant symptoms are rare after one-sided adrenal gland RT.
- Chemotherapy may have influenced the results (e.g., Cisplatin has been reported to induce Corticosteroid release in an animal model, Aggarwal et al., Anticancer Drugs 1994; furthermore, it has been reported to be associated with decreased cortisol levels after application in patients: Morrow et al., 2002, psychophysiology); whichever effect.
- Corticosteroids which are typically applied in conjunction with chemotherapy and are used in many patients treated with RT may have influenced the results (Yeung et al., 1998, Endocrine Reviews); did all three patients receive corticosteroids to mitigate anticancer treatment effects or only the two (of three) chemotherapy patients?
Responses
- Thanks for your great comments. In our review, there are no literatures discussing about the adequate time frames of compensation of decreased cortisol induced by irradiation in mice. This issue is still under investigation, and listed as a limitation in Discussion as follows: “A third limitation was the timing and duration of AIR-induced biological effects may need a precise and long-term follow-up for evaluation in a large-scale study.”
- We only had a short-term follow up because we intended to research the acute adverse effects of irradiation to adrenal gland in this study. The swimming endurance test was performed after wound healing around 5 days after surgery and RT. Then blood sampling was performed at 15:00 and 20:00 on the 6th day. Histopathology of the adrenal gland was sampled on the 7th day. The time between the last blood sampling and histopathological sampling was around 12 hours. The issue of adequate timing for blood and histopathological sampling is still under investigation, and this limitation is noted in Discussion as follows: “A third limitation was the timing and duration of AIR-induced biological effects may need a precise and long-term follow-up for evaluation in a large-scale study.”
- Thanks for your suggestion of a further discussion. There were several literatures discussing about the relationship of RT-induced fatigue and systemic TNF-α which induced by localized RT in normal tissues. We’ve added an additional description in Discussion as follows: “In addition to the direct RT damage on adrenal gland, localized RT may produce systemic effects via inflammatory responses. There were several literatures discussing about systemic tumor necrosis factor alpha (TNF-α) induced by localized RT in normal tissues. The release of intracellular molecules from RT-induced injured cells would trigger innate immune responses characterized by the upregulation of pro-inflammatory cytokines such as TNF-α (McDonald et al., 2016, Radiat Res.; Villar et al., 2013, PLoS One.; Judd et al., 2000, Ann N Y Acad Sci.). Previous studies have shown that TNF-α inhibited basal and ACTH-stimulated cortisol secretion (Jäättelä et al., 1991, Endocrinology.; Vandermeer et al., 1996, Cytokine.). Also, RT-induced fatigue was associated with increased pro-inflammatory gene expression in normal tissues such as liver TNF-α mRNA and in circulation (McDonald et al., 2016. Radiat Res.)”
- As our review in Discussion, most radiation-induced adrenal toxicities were grade I or II toxicities, and the evaluation of toxicity profiles was mostly from RT for adrenal metastases. We regarded healthy adrenal glands, but not metastatic ones, as essential organs to spare in RT planning, which differed from previous studies. Therefore, the toxicities should be the lower the better for the incidental RT to adrenal gland.
- Thanks for your suggestion. Chemotherapy may influence the results. Our study focused on the radiation effects on adrenal gland. This is a limitation, and shown in Discussion as follows: “Finally, other possible etiologies such as chemotherapy that cause fatigue should be examined in future work.”
- Two of these patients received CCRT with corticosteroids as standard treatment. These 2 patients received weekly chemotherapy, and a single low dose of 5 mg dexamethasone was used in each course of chemotherapy.

Reviewer 3 Report
The authors should be congratulated for an interesting and fairly well designed experiment to investigate this question. Fatigue is the most common side effect of radiotherapy. What contribution, if any, of moderate to high dose radiation, or of low dose spill, to the adrenals, to this common side effect, is an intriguing question and a reasonable hypothesis that merits investigation.
My comments are as follows:
1) The experimental inclusion of a sham group and 0Gy group are a good design. The remaining condition I would question is whether there needs to be a 2Gy non-adrenal arm, i.e 2Gy of RT to another, similar target, to rule out off target effects of radiation, for example inflammatory response or otherwise.
2) While the authors correctly identify several clinical studies reporting a handful of cases of Grade 1-2 adrenal insufficiency in patients treated with mostly SBRT for adrenal metastases, this must be placed in context of well over a thousand patients in the cumulative literature treated with adrenal SBRT. The rate of clinically apparent adrenal insufficiency significant enough to be detected on routine clinical follow up is apparently likely less than 1%, despite high doses of SBRT to adrenal metastases -- implying high doses of RT to the adrenal gland, much higher than in the author's experimental model. The author's should speculate on why this might be.
3) Even unilateral surgical adrenalectomy rarely leads to adrenal insufficiency due to compensation of the contralateral adrenal. How, then, do the author's interpret their mouse experimental data of unilateral irradiation? Are the cortisol changes a transient change which may recover in the longer term with compensation from the remaining adrenal? Are they a transient stress response to an additional stressor (radiation)? How can we be sure this stress response is specific to adrenal RT versus RT in general (lack of a 2Gy non-adrenal arm)?
4) Why was ACTH not measured in the mice to distinguish between central vs peripheral effects?
5) 4 mice per arm is rather low and is a possible limitation of this study
6) Regarding the clinical data of the 3 patients, the authors should discuss more the limitations of this correlative data. For example, 2 of 3 patients received concurrent chemotherapy, which could just as easily be conceived to affect cortisol and HPA axis as a stressor. A comparison group of patients undergoing similar doses of RT for whom the adrenals were NOT in the field would make for an interesting comparison. Yet I suspect such patients (rectal cancer? head and neck?) may have similar changes in cortisol due to general stress, concurrent chemotherapy, etc.
7) It is hard to reconcile the histologic report of radiation injury at a low dose of 2Gy, with the reputation of the adrenal gland as a relatively radiation resistant organ. Histologic evidence of radiation injury to the adrenal gland appears to be lacking in the literature. Are the authors aware of any other studies reporting histologic changes at such a low dose, or any dose of RT? The study would be strengthened by establishing a dose response, and better description of variation and extent of histologic changes in the adrenals post-RT, and more paired images of ipsilateral/contralateral adrenals (perhaps in a supplement).
Author Response
Thanks for your suggestion for this article. We have made some point-by-point responses to the comments of reviewer.
Comments of reviewer
- The experimental inclusion of a sham group and 0 Gy group are a good design. The remaining condition I would question is whether there needs to be a 2 Gy non-adrenal arm, i.e 2 Gy of RT to another, similar target, to rule out off target effects of radiation, for example inflammatory response or otherwise.
- While the authors correctly identify several clinical studies reporting a handful of cases of Grade 1-2 adrenal insufficiency in patients treated with mostly SBRT for adrenal metastases, this must be placed in context of well over a thousand patients in the cumulative literature treated with adrenal SBRT. The rate of clinically apparent adrenal insufficiency significant enough to be detected on routine clinical follow up is apparently likely less than 1%, despite high doses of SBRT to adrenal metastases -- implying high doses of RT to the adrenal gland, much higher than in the author's experimental model. The author's should speculate on why this might be.
- Even unilateral surgical adrenalectomy rarely leads to adrenal insufficiency due to compensation of the contralateral adrenal. How, then, do the author's interpret their mouse experimental data of unilateral irradiation? Are the cortisol changes a transient change which may recover in the longer term with compensation from the remaining adrenal? Are they a transient stress response to an additional stressor (radiation)? How can we be sure this stress response is specific to adrenal RT versus RT in general (lack of a 2 Gy non-adrenal arm)?
- Why was ACTH not measured in the mice to distinguish between central vs peripheral effects?
- 4 mice per arm is rather low and is a possible limitation of this study.
- Regarding the clinical data of the 3 patients, the authors should discuss more the limitations of this correlative data. For example, 2 of 3 patients received concurrent chemotherapy, which could just as easily be conceived to affect cortisol and HPA axis as a stressor. A comparison group of patients undergoing similar doses of RT for whom the adrenals were NOT in the field would make for an interesting comparison. Yet I suspect such patients (rectal cancer? head and neck?) may have similar changes in cortisol due to general stress, concurrent chemotherapy, etc.
- It is hard to reconcile the histologic report of radiation injury at a low dose of 2 Gy, with the reputation of the adrenal gland as a relatively radiation resistant organ. Histologic evidence of radiation injury to the adrenal gland appears to be lacking in the literature. Are the authors aware of any other studies reporting histologic changes at such a low dose, or any dose of RT? The study would be strengthened by establishing a dose response, and better description of variation and extent of histologic changes in the adrenals post-RT, and more paired images of ipsilateral/contralateral adrenals (perhaps in a supplement).
Responses
- Thanks for your suggestion of a further discussion. There were several literatures discussing about the relationship of RT-induced fatigue and systemic TNF-α which induced by localized RT in normal tissues. We’ve added an additional description in Discussion as follows: “In addition to the direct RT damage on adrenal gland, localized RT may produce systemic effects via inflammatory responses. There were several literatures discussing about systemic tumor necrosis factor alpha (TNF-α) induced by localized RT in normal tissues. The release of intracellular molecules from RT-induced injured cells would trigger innate immune responses characterized by the upregulation of pro-inflammatory cytokines such as TNF-α (McDonald et al., 2016, Radiat Res.; Villar et al., 2013, PLoS One.; Judd et al., 2000, Ann N Y Acad Sci.). Previous studies have shown that TNF-α inhibited basal and ACTH-stimulated cortisol secretion (Jäättelä et al., 1991, Endocrinology.; Vandermeer et al., 1996, Cytokine.). Also, RT-induced fatigue was associated with increased pro-inflammatory gene expression in normal tissues such as liver TNF-α mRNA and in circulation (McDonald et al., 2016. Radiat Res.)”
- Thanks for your suggestion of a further discussion. There were no literatures to discuss the effects of radiation on adrenal gland in an animal study. The intrinsic radiosensitivity of mice may differ from human in various tissues, and the body weight and body shape of mice and human were also different. The actual effects of 2 Gy on adrenal gland in mice were evaluated in this study, and the results might be underestimated before. Dose‐response analysis might be useful for this issue, and it was a limitation disclosed in Discussion as follows: “Dose‐response analysis may be performed if more study population included.”
- We only had a short-term follow up because we intended to research the acute adverse effects of irradiation to adrenal gland in this study, including animal model and clinical observation. We appreciate the helpful comment on the possible long-term compensation of adrenal insufficiency. This point has been added in limitation of study in Discussion as follows: “A third limitation was the timing and duration of AIR-induced biological effects may need a precise and long-term follow-up for evaluation in a large-scale study.”
For the possible off target effects, we’ve added an additional description in Discussion as follows: “In addition to the direct RT damage on adrenal gland, localized RT may produce systemic effects via inflammatory responses. There were several literatures discussing about systemic tumor necrosis factor alpha (TNF-α) induced by localized RT in normal tissues. The release of intracellular molecules from RT-induced injured cells would trigger innate immune responses characterized by the upregulation of pro-inflammatory cytokines such as TNF-α (McDonald et al., 2016, Radiat Res.; Villar et al., 2013, PLoS One.; Judd et al., 2000, Ann N Y Acad Sci.). Previous studies have shown that TNF-α inhibited basal and ACTH-stimulated cortisol secretion (Jäättelä et al., 1991, Endocrinology.; Vandermeer et al., 1996, Cytokine.). Also, RT-induced fatigue was associated with increased pro-inflammatory gene expression in normal tissues such as liver TNF-α mRNA and in circulation (McDonald et al., 2016. Radiat Res.)”
- The blood samples were collected via retro-orbital blood sampling at 15:00 and 20:00 on the sampling day to represent baseline and evening cortisol levels. For the little volume of each sampling, we only analyzed cortisol levels in this study.
- 4 mice per arm is a limitation of this study, and we’ve followed the 3Rs principle to reduce the number of animals used in this study. The limitation statement is revised in Discussion as follows.
Previous version:
“Secondly, the number of patients in our clinical observation with assessable serum cortisol and ACTH levels was too small to draw a firm conclusion.”
Revised version:
“Secondly, the number of patients in our clinical observation with assessable serum cortisol and ACTH levels was too small to draw a firm conclusion, and the number of mice in our animal study was also too small.”
- Thanks for your practical comments. In limitation part, we’ve illustrated that other possible etiologies including chemotherapy would cause fatigue. Our study focused on the radiation effects on adrenal gland. This was described in Discussion as follows: “Finally, other possible etiologies such as chemotherapy that cause fatigue should be examined in future work.”
- In our review, there were no literatures discussing the histological changes of adrenal gland after radiation therapy in an animal study. It might be our mistake to use “characteristic RT injury features” for interpretation. To avoid confusion, we have eliminated the use of “characteristic” in Results. No dose‐response analysis was a limitation depicted in Discussion as follows: “Dose‐response analysis may be performed if more study population included.”

Round 2
Reviewer 2 Report
Thank you for revising the manuscript. Several issues have been resolved; however, I think there are a few points remaining which should be adressed:
While the authors acknowledged that some patients received corticosteroids weekly in the response to my question, they do not mention this weakness in the paper. It would be relevant that the authors analyze the potential impact of high-dose corticosteroids on their results. Depending on the timing of blood tests, this might have relevantly influenced the results.
Furthermore, the sentence “Finally, other possible etiologies such as chemotherapy that cause fatigue should be examined in future work.” is not enough. The authors must clearly state that chemotherapy might have influenced their results and that it is unclear in which parts the results of the patient study can be attributed to radiation therapy and chemotherapy.
The authors added a section on TNF-α which is well-written but fails to acknowledge that a) surgical series showed that only minor rests of adrenal tissue are required to preserve hormonal function and that b) multiple series of radiation therapy showed that irradiation of adrenal lesions is safe and effective. Instead they detailed laboratory findings about potential off-target effects of radiation therapy; however, the clinical implications of these findings are unclear. If the authors would like to proof such effects, it would be helpful to have an out-of-field radiation therapy control group.
Finally, the conclusion of the abstract is not supported by the data and should be rephrased.
Author Response
Thanks for your suggestion for this article. We have made some point-by-point responses to the comments of reviewers.
Comments
- While the authors acknowledged that some patients received corticosteroids weekly in the response to my question, they do not mention this weakness in the paper. It would be relevant that the authors analyze the potential impact of high-dose corticosteroids on their results. Depending on the timing of blood tests, this might have relevantly influenced the results.
- Furthermore, the sentence “Finally, other possible etiologies such as chemotherapy that cause fatigue should be examined in future work.” is not enough. The authors must clearly state that chemotherapy might have influenced their results and that it is unclear in which parts the results of the patient study can be attributed to radiation therapy and chemotherapy.
- The authors added a section on TNF-α which is well-written but fails to acknowledge that a) surgical series showed that only minor rests of adrenal tissue are required to preserve hormonal function and that b) multiple series of radiation therapy showed that irradiation of adrenal lesions is safe and effective. Instead they detailed laboratory findings about potential off-target effects of radiation therapy; however, the clinical implications of these findings are unclear. If the authors would like to proof such effects, it would be helpful to have an out-of-field radiation therapy control group.
- Finally, the conclusion of the abstract is not supported by the data and should be rephrased.
Responses
- Thanks for your suggestion of a further discussion. There are several literatures discussing about the potential impact of corticosteroids on adrenal function. We’ve added an additional description in Discussion as follows: “Corticosteroids have been widely-used with chemotherapy for prophylaxis and treatment of chemotherapy-related symptoms, and may influence the adrenal function (Yeung et al., 1998, Endocr Rev). The impact of corticosteroids on adrenal function has been proved to be dose and duration dependant. Use of corticosteroids in low or high doses resulted in a percentage of adrenal insufficiency of 2.4 and 21.5%, respectively, and short or long term use resulted in a percentage of adrenal insufficiency of 1.4% and 27.4%, respectively (Broersen et al., 2015, J Clin Endocrinol Metab). In this study, two patients received CCRT with single low-dose of corticosteroids (dexamethasone 4 mg via infusion) weekly prior to chemotherapy. If corticosteroids were administered at high doses, the serum cortisol levels might be suppressed within 24-48 hours, recovering after 1-4 weeks (Habib et al, 2009, Clin Rheumatol). Therefore, use of corticosteroids might have relevantly influenced the results, but the impact might be relatively unremarkable in short-term low-dose administration.”
- Thanks for your great comments. Chemotherapy may influence the results. For this issue, we’ve added an additional description in Discussion as follows: “Chemotherapy may influence the study results. Cisplatin has been reported to induce corticosteroid release in an animal model (Aggarwal et al., 1994, Anticancer Drugs) and it has been reported to be associated with decreased cortisol levels in human (Morrow et al., 2002, Psychophysiology). There were several literatures discussing about chemotherapy-induced adrenal insufficiency in cancer patients (Cornell et al., 2013, Case Rep Endocrinol; Kim et al., 2016, Oncology). For patient treated with CCRT, the adrenal insufficiency might be attributed to RT and chemotherapy. The impacts of RT and chemotherapy should be evaluated separately, and may need further investigation.”
- Thanks for your great comments to clarify these issues. a) Only minor rests of adrenal tissue are required to preserve adrenal function in surgical series. We’ve added an additional description in Discussion as follows. “Surgical series indicated that after complete removal of one adrenal gland, only a minority of patients developed clinically relevant adrenal insufficiency (Mitchell et al., 2009, Surg Endosc). The extent, timing, and duration of clinical symptoms and hormone alterations induced by surgery and adrenal RT might be different, and the regulation of HPA axis could be distinct. In our study, RT-induced adrenal insufficiency was observed. Further research would be required for this issue, and RT-induced direct adrenal damages and TNF-α production that inhibited basal and ACTH-stimulated cortisol secretion may be possible causes.” b) We’ve added an additional description in Discussion as follows. “Various multicenter series on patients who received RT to adrenal glands have been published in recent years. Some series reported with detailed toxicity profiles, including adrenal insufficiency after one-sided or two-sided RT. Most radiation-induced adrenal toxicities were grade I or II (Buergy et al., 2021, Int J Cancer). There was sparse literature considering the adrenal glands as organs at risk in RT. The evaluation of toxicity profiles was mostly from SBRT for adrenal metastases. We regarded healthy adrenal glands, but not metastatic ones, as essential organs to spare in RT planning, which differed from previous studies. Therefore, the toxicities should be the lower the better for the incidental RT dose to adrenal gland.” Lack of a non-adrenal irradiation control group is a limitation of this study. For this issue, we’ve added some descriptions in Discussion as follows. “In addition to the direct RT damage on adrenal gland, localized RT may produce systemic effects via inflammatory responses. There were several literatures discussing about systemic tumor necrosis factor alpha (TNF-α) induced by localized RT in normal tissues. The release of intracellular molecules from RT-induced injured cells would trigger innate immune responses characterized by the upregulation of pro-inflammatory cytokines such as TNF-α. Previous studies have shown that TNF-α inhibited basal and ACTH-stimulated cortisol secretion. Also, RT-induced fatigue was associated with increased pro-inflammatory gene expression in normal tissues such as liver TNF-α mRNA and in circulation. Therefore, RT effects on non-adrenal tissue might influence adrenal function, and should be examined in the further study.” “Fourthly, a non-adrenal irradiation control group may help to examine the effects of RT to normal tissue on adrenal insufficiency.”
- Thanks for your suggestion. The conclusion of the abstract is revised as follows: “The RT dose distributed to adrenal gland may have correlation to unwanted adverse effects including fatigue and adrenal hormone alterations.”

Reviewer 3 Report
Thank you for addressing some of my concerns. Please help address the following:
1) In addition to the new discussion on off target effects due to inflammatory response, please also discuss the limitation of the lack of a non-adrenal 2Gy irradiation control group in the study design.
2) Please discuss specifically that interpretation of the human data is limited by concurrent chemotherapy.
3) Please provide further histologic slide examples of RT-injured adrenal cortex alongside normal cortex as a supplement. Because there are only 4 mice, it should be feasible to provide images for all mice.
4) Please soften the conclusion of the abstract to match the tone of the main body conclusion, i.e that the data presented are hypothesis generating.
5) Please discuss the limitation of not measuring ACTH levels in mice to establish etiology of cortisol decrease as peripheral rather than central.
Author Response
Thanks for your suggestion for this article. We have made some point-by-point responses to the comments of reviewers.
Comments
- In addition to the new discussion on off target effects due to inflammatory response, please also discuss the limitation of the lack of a non-adrenal 2Gy irradiation control group in the study design.
- Please discuss specifically that interpretation of the human data is limited by concurrent chemotherapy.
- Please provide further histologic slide examples of RT-injured adrenal cortex alongside normal cortex as a supplement. Because there are only 4 mice, it should be feasible to provide images for all mice.
- Please soften the conclusion of the abstract to match the tone of the main body conclusion, i.e that the data presented are hypothesis generating.
- Please discuss the limitation of not measuring ACTH levels in mice to establish etiology of cortisol decrease as peripheral rather than central.
Responses
- Thanks for your great comment. Lack of a non-adrenal irradiation control group is a limitation of this study. For this issue, we’ve added some descriptions in Discussion as follows. “In addition to the direct RT damage on adrenal gland, localized RT may produce systemic effects via inflammatory responses. There were several literatures discussing about systemic tumor necrosis factor alpha (TNF-α) induced by localized RT in normal tissues. The release of intracellular molecules from RT-induced injured cells would trigger innate immune responses characterized by the upregulation of pro-inflammatory cytokines such as TNF-α. Previous studies have shown that TNF-α inhibited basal and ACTH-stimulated cortisol secretion. Also, RT-induced fatigue was associated with increased pro-inflammatory gene expression in normal tissues such as liver TNF-α mRNA and in circulation. Therefore, RT effects on non-adrenal tissue might influence adrenal function, and should be examined in the further study.” “Fourthly, a non-adrenal irradiation control group may help to examine the effects of RT to normal tissue on adrenal insufficiency.”
- Thanks for your great comments. Chemotherapy may influence the results. For this issue, we’ve added an additional description in Discussion as follows: “Chemotherapy may influence the study results. Cisplatin has been reported to induce corticosteroid release in an animal model (Aggarwal et al., 1994, Anticancer Drugs) and it has been reported to be associated with decreased cortisol levels in human (Morrow et al., 2002, Psychophysiology). There were several literatures discussing about chemotherapy-induced adrenal insufficiency in cancer patients (Cornell et al., 2013, Case Rep Endocrinol; Kim et al., 2016, Oncology). For patient treated with CCRT, the adrenal insufficiency might be attributed to RT and chemotherapy. The impacts of RT and chemotherapy should be evaluated separately, and may need further investigation.”
- Thanks for your practical comments. The histopathology of adrenal cortex from each mouse of the 2 Gy group is provided in the supplement. We’ve added an additional description in Results as follows: “Histopathology of the unirradiated right adrenal cortex and RT-injured left adrenal cortex from each mouse of the 2 Gy group was illustrated in Supplementary figure 1.”
- Thanks for your suggestion. The conclusion of the abstract is revised as follows: “The RT dose distributed to adrenal gland may have correlation to unwanted adverse effects including fatigue and adrenal hormone alterations.”
- This is a limitation that ACTH is not measured in the animal study. We’ve added an additional description in Discussion as follows: “Fifthly, ACTH was not measured in the animal study to distinguish between central or peripheral adrenal insufficiency. The blood samples were collected via retro-orbital blood sampling at 15:00 and 20:00 on the sampling day to represent baseline and evening cortisol levels. For the little volume of each sampling, we only analyzed cortisol levels in this study.”
